# Exploring areas of consensus and conflict around values underpinning public involvement in health and social care research: a modified Delphi study

D Snape,[1] J Kirkham,[2] J Preston,[3] J Popay,[4] N Britten,[5] M Collins,[4] K Froggatt,[4] A Gibson,[5] F Lobban,[4] K Wyatt,[5] A Jacoby[1]

For numbered affiliations see end of article.

**Correspondence to**
Professor Ann Jacoby;
ajacoby@liv.ac.uk

## ABSTRACT

**Objective:** There is growing interest in the potential benefits of public involvement (PI) in health and social care research. However, there has been little examination of values underpinning PI or how these values might differ for different groups with an interest in PI in the research process. We aimed to explore areas of consensus and conflict around normative, substantive and process-related values underpinning PI.

**Design:** Mixed method, three-phase, modified Delphi study, conducted as part of a larger multiphase project.

**Setting:** The UK health and social care research community.

**Participants:** Stakeholders in PI in research, defined as: clinical and non-clinical academics, members of the public, research managers, commissioners and funders; identified via research networks, online searches and a literature review.

**Results:** We identified high levels of consensus for many normative, substantive and process-related issues. However, there were also areas of conflict in relation to issues of bias and representativeness, and around whether the purpose of PI in health and social care research is to bring about service change or generate new knowledge. There were large differences by group in the percentages endorsing the ethical justification for PI and the argument that PI equalises power imbalances. With regard to practical implementation of PI, research support infrastructures were reported as lacking. Participants reported shortcomings in the uptake and practice of PI. Embedding PI practice and evaluation in research study designs was seen as fundamental to strengthening the evidence base.

**Conclusions:** Our findings highlight the extent to which PI is already embedded in research. However, they also highlight a need for 'best practice' standards to assist research teams to understand, implement and evaluate PI. These findings have been used in developing a Public Involvement Impact Assessment Framework (PiiAF), which offers guidance to researchers and members of the public involved in the PI process.

### Strengths and limitations of this study

- Despite growing interest in the UK and internationally in the potential benefits of public involvement (PI) in research, there has been little examination of the values underpinning it, or of how these values might be different for different stakeholders. As part of a large study developing guidance on how to assess the impact of PI in research, we undertook a mixed method, modified Delphi study which has provided primary evidence of areas of consensus and conflict around the value systems underpinning PI in research.
- Our mixed method, modified Delphi study involved a heterogeneous panel of PI experts, reflective of the range of key stakeholder groups and was geographically diverse.
- 'Consensus' thresholds were determined in advance of data collection, with two levels of consensus defined.
- A limitation of our study was that panellists likely were individuals with a strong commitment to PI; we can therefore say nothing about the values of those who choose not to engage with the PI enterprise. A further limitation is that response rates to the two-round survey were low and that our conclusions are potentially biased. However, a large percentage of the round 1 survey responders also completed the round 2 survey; and the quality of the data was high.
- Our findings not only provide important evidence about value consensus, but also highlight a need for 'best practice' standards to assist research teams in the PI process.

## INTRODUCTION

There is a growing interest internationally in the potential benefits of public involvement (PI) in health research.[1–3] Within the UK, the involvement of members of the public in research is firmly established in health and social care policy,[4] and organisations including, for example, the National Institute for

Health Research[5] increasingly identify PI as a prerequisite for funding.

There is a plethora of literature illuminating the experiences of members of the public and professional researchers, and the process of PI and its potential impacts.[6–9] However, while studies report on use of PI in context of priority setting,[10] [11] the conduct of clinical trials[12] and the identification of treatment outcomes,[13] there has been relatively little examination of the values underpinning PI or about capturing and assessing the impact of PI effectively.[14–18] Possible reasons cited for the limited number of studies evaluating the impact of PI are that evaluation is too difficult and that PI is of intrinsic value and as such needs no further justification.[16] [19–21] The authors of the present article acknowledge this latter perspective. However, we would argue that the current lack of a research evidence base around the impact of PI presents a challenge, since without such an evidence base it is difficult to ensure integrity of the PI endeavour, avoid potential adverse effects and ensure that it is adequately resourced. The modified Delphi study reported here was part of a larger Medical Research Council funded study (G0902155/93948) which aimed to review evidence on the values and impacts associated with PI, develop guidance on how these impacts can be assessed and contribute to the development of good practice standards for PI. The modified Delphi study utilised mixed methods to explore areas of consensus and conflict underpinning three over-arching value systems identified from a preceding literature review.[22] These values relate to (1) normative perspectives on PI, which consider involvement as an end in itself, for example, rights and empowerment; (2) substantive perspectives which consider the consequences of PI, for example, quality and relevance and (3) process-related values associated with good involvement, for example, partnership and equality.

## AIM AND OBJECTIVES OF THE DELPHI STUDY

The overall aim of the modified Delphi Study was to explore areas of consensus and conflict around the normative, substantive and process-related values underpinning PI in health and social care research. The specific objectives were:

▶ To explore key issues identified from our literature review with PI experts;
▶ To provide new knowledge about the views of different stakeholders on PI;
▶ To contribute to the development of guidance on how to assess the impact of PI in research.

## METHODS

In this study, we defined PI as an active partnership between members of the public and researchers in the research process, rather than the use of people as the 'subjects' of research. The term 'public' has been used as developed by the UK National Advisory Group

INVOLVE,[23] and includes patients and potential patients, carers and people who use health and social care services. Within the study reported here, PI has been achieved through the collaboration of academic researchers with experience of working in the field of PI, user investigators and the wider project's Public Advisory Group whose members contributed to all five phases of the work. A mixed methods approach to data collection was employed in a three-stage, modified Delphi process:

▶ Three expert workshops (participant total n=42), including members of the public, academic and user-researchers, researcher/clinicians, research funders and research managers were conducted to generate qualitative data. The range of normative, substantive and process-related values underpinning PI, identified in a previously conducted literature review,[22] were discussed, to identify concepts to be explored at round 1 (R1) and round 2 (R2) of the subsequent modified Delphi survey;
▶ A pilot study to test the R1 survey questionnaire. As a strategy to reduce attrition,[24] [25] careful attention was paid to the content and layout of the invitation email, the survey layout and the clarity of questions. Piloting was conducted with academic (n=6) and user-researchers (n=3) and members of our Patient Advisory Group (n=2). Language, question type and questionnaire formatting were edited in response to participant feedback;
▶ A two-round modified Delphi survey was undertaken between November 2011 and September 2012 to explore areas of consensus and conflict around the normative, substantive and process-related values underpinning PI in health and social care research.

In this article, we describe the modified Delphi process and report on the findings relating to the values underpinning PI in health and social care research. A subsequent article will report the survey findings in relation to perceived PI barriers and facilitators, perceived impacts of PI and potential approaches to its evaluation.

### A modified Delphi technique

The main premise of the Delphi technique is the assumption that group opinion is more valid than individual opinion. The technique offers a reliable data collection method in circumstances where there is uncertainty or a paucity of knowledge surrounding the topic area under investigation.[26–30] While the Delphi technique has been used extensively within health and social science research,[31–38] criticism of the technique includes the perception that it seeks to force consensus and so is weakened by not allowing panellists to elaborate on their views.[32] In response to this criticism, the current study used a modified Delphi technique which did not force consensus; rather panellists were provided with opportunities to elaborate on why they held the views they expressed or endorsed,[33] in order to allow us

to try and tease out areas of conflict as well as areas of consensus.

There are a number of key concepts and assumptions which distinguish the basic Delphi technique from other research methods. These include anonymity, multistage iteration and controlled feedback, exploration of consensus via statistical group response and the use of experts.[30] [39] Each of these key characteristics was given due consideration as a means of enhancing the validity and reliability of the study design and the quality of responses.[28] [40–43]

## Sampling and the use of 'experts'

To reduce possible bias, our sampling strategy for panel composition was reflective of the population subgroups under investigation and geographically diverse.[31] [33] To increase the reliability of the results, a large heterogeneous panel of 'experts', consisting of distinct stakeholder groups was recruited.[24] We defined 'experts' as a group of *informed individuals*[28] [44] [45]—which, in this instance, was knowledge and/or experience of PI in health and social care research. Sampling was purposive. We aimed to capture the various PI perspectives and interests of members of the public, user, academic and clinical researchers and research managers, directors, commissioners and funders. Potential panellists were identified in one of three ways: directly, through research team members' contacts and networks; through online searches of open-access research information and funding sites and via a review of literature in the field of PI in health and social care research.

## Anonymity

Complete anonymity could not be achieved as survey panellists and their responses were known to some members of the research team. However, anonymity between panellists was guaranteed. Reactions to panellists' opinions, key arguments and levels of consensus for each subgroup were fed back to panellists at R2 of the modified Delphi survey; each opinion carried the same weight and was afforded the same degree of importance in the analysis. In this way, participant bias was eliminated.[33] This approach also enabled panellists to be open and honest about their views on various issues as well as providing them with an equal opportunity to express an opinion without feeling pressured psychologically to conform to the views of others.[33]

## Quantitative data analysis

Determining how consensus cut-off points are reached and understanding how they are derived are often problematic, as the reporting of such criteria is limited within many published studies.[29] To address this, we sought statistical advice and defined specific criteria for determining consensus thresholds a priori.

We took, as representing clear consensus, endorsement or rejection of a statement by at least 60% of respondents. However, attention was also paid to the distribution of responses of the remaining 40%. Where these responses clustered in one response option only, consensus was not assumed and this item was further explored in R2 of the survey. We took, as representing critical consensus, endorsement or rejection of a statement by at least 70% of respondents. An additional criterion for critical consensus was that at least 55% of responses either endorsed or rejected the extreme categories in the scales.

At R2, the results from the R1 survey were fed back to panellists in the form of bar charts, textual summaries of statistical data and summarised descriptions of qualitative findings. Feedback included comparison of the responses of the different subgroups for each item. Particular attention was drawn to items demonstrating either within-subgroup or between-subgroup dissent. Items at R1 where between-subgroup differences were greater than 10% were further explored at R2. Also explored at R2 were any 'unexpected' (in the collective view of the study team) endorsement/rejection of items by the subgroups. We draw attention to these 'unexpected' responses in the findings section below.

## Qualitative data analysis

In R1 and R2, to enable in-depth exploration of the quantitative findings, panellists were provided with the opportunity to make further comments as they saw fit, through open questions. Thematic codes were identified using Framework Analysis,[46] a matrix-based method for ordering and synthesising data. The analysis was conducted by DS. Quality checking of the coding process and reduction of coding bias were ensured by AJ, who reviewed 10% of the qualitative data. First, data were reviewed inductively to identify recurring themes and concepts raised by panellists. These were coded and formed the initial major and subthemes; additional codes were then incorporated through an iterative process involving DS and AJ. The thematic framework was further refined before being applied systematically to the whole dataset. This process facilitated the identification of any inconsistencies in coding, which were subsequently discussed and reconciled.

## RESULTS

The iterative nature of the Delphi technique enabled R2 of the survey to be informed by the results of R1. For this reason, the methods and results for both rounds are discussed together.

## Panellists

At R1, 740 (n=740) potential 'expert' panellists were invited, via email, to participate in the online survey. Non-responders were emailed two reminder letters as appropriate, yielding a total response of 318 (n=318) (response rate 43%). Responding panellists self-selected themselves into one of five 'stakeholder' groups: clinical academic (CA), (20%); non-CA (NCA), (28%); member

**Table 1**  Research experience by stakeholder group*

| Stakeholder group | n | Minimum 5 years research experience | Some PI responsibility | Formal training relevant to PI |
|---|---|---|---|---|
| Clinical academic researcher | 63 | 52 (82.5%) | 52 (82.5%) | 27 (42.9%) |
| Non-clinical academic researcher | 88 | 70 (79.5%) | 63 (71.6%) | 27 (30.7%) |
| Member of the public | 55 | 33 (60.0%) | 27 (49.1%) | 35 (63.6%) |
| Research manager/funder | 76 | 53 (69.7%) | 64 (84.2%) | 31 (40.8%) |
| Dual role | 34 | 30 (88.2%) | 29 (85.3%) | 14 (41.2%) |

*Data taken from R1.
PI, public involvement; R1, round 1.

of the public (MP), (17%); research manager or funding/commissioning body employee (RM), (24%); or occupying multiple roles (MR), (11%).

The characteristics of those who took part in R1 are shown in table 1. Overall, the participants had high levels of expertise, with most of them reporting having at least 5 years of experience in research and three-quarters having some responsibility for PI. However, there were also differences in level of expertise, members of the public being least likely to have lengthy research experience or any specific PPI responsibilities. In contrast, though fewer than half (n=134; 48%) of all the panellists reported having had any formal training relevant to PI, members of the public were far more likely to have had such training (table 1), while academic researchers tended to emphasise 'learned by doing'.

The panellists reported being involved in a wide range of research roles including, for example, drafting research proposals and protocols (n=239; 86%); developing plain English summaries (n=238; 86%); sitting on project advisory and steering groups (n=228; 82%); presenting research findings (n=226; 82%) and being involved in data collection (n=223; 815). The panellists cited a number of practical learning opportunities which underpinned their PI skills acquisition, including, for example:

Good networking. [NCA, R1]

"…hard work building rapport and understanding of the patients' perspective, putting myself in their shoes. [NCA, R1]

Support from others, learning alongside others, and time. [CA, R1]

…so learning from mistakes and trying to listen to what both the patients and the researchers need to make it a meaningful experience. [MP, R1]

Various policy documents for engaging the public in research on which we can rest our local framework and priorities. [RF, R1]

All the panellists (n=318; response rate 43%) who submitted the R1 questionnaire were eligible to participate in R2. Of the 318 responders, three electronically 'opted out' of receiving further email communications. The R2 questionnaire was therefore emailed to 315 (n=315) panellists. Up to two reminders were actioned as required. A total of 231 (n=231) R2 survey responses were received across the five stakeholder groups: CA, (17%); NCA, (29%); MP, (18%); RM, (24%) and MR), (12%), yielding a R2 response rate of 73% (of the 43% who took part in R1).

### Arguments related to the nature of knowledge

We asked our panellists to comment on what kinds of knowledge they believed should inform health and social care research. Responses at R1 (see figure 1) revealed critical consensus across the different stakeholder groups (over 90% strongly agreed/agreed) for the substantive argument that members of the public have unique knowledge and expertise that is complementary to that of professionals/clinicians and researchers, and should be valued equally.

A comment from one CA panellist illustrates this:

It is important not to conflate the complementary perspectives of clinical/health, social care professional researchers versus methodologists. Both have important knowledge and expertise to bring to the table in informing health and social care research, as do patients, their informal carers and advocates and members of the public…it is important that all of these perspectives are regarded as equally important, rather than privileging any one view over the others. [CA; R1]

Though it just failed to reach our definition of clear consensus, there was broad support across the stakeholder groups (58%) for the idea that while professionals/clinicians and researchers may also be service users, they cannot themselves represent user issues effectively. Since there was also some support across stakeholder groups (just over 50% strongly agreed/agreed) for the idea that members of the research community and members of the public are likely to have biased views about research, we explored this issue of bias further in R2.

At R2, we asked panellists to comment on whether they felt it mattered if different stakeholder groups held views which others considered biased. Forty-three per

**Figure 1** What kinds of knowledge should inform health and social care research? Data taken from Round 1.

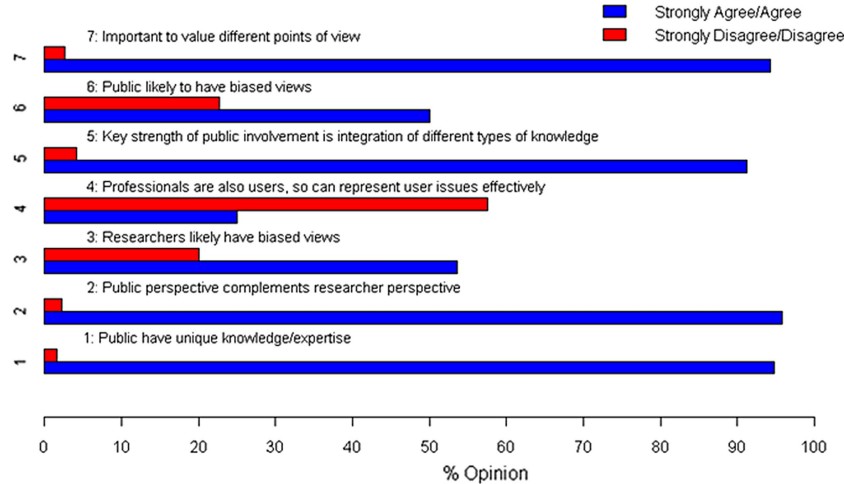

cent said 'yes' they did feel it mattered, with a percentage range of 29–59% across stakeholder groups. Individual stakeholder group responses are outlined in table 2.

A common theme across panellists' written comments and one articulated by a panellist holding multiple roles was that all research actors are biased to some degree, holding different views based on knowledge and/or experience.

> All views will have a bias of some kind, but I think it is more important that the view is 'informed'. Bringing together the different perspectives in an open, collaborative way is one of the strengths of patient involvement, not a weakness. [MR, R2].

Many panellists articulated that differences in perspective should not necessarily be viewed negatively; rather they should be acknowledged as a 'given'. Interestingly, at R2, the issue of power (who holds the power to enforce their viewpoint) was highlighted as more important than the question of bias.

Following from this, in R2 process, values that involve 'building relationships of trust between the different stakeholder groups' [NCA, R2] were frequently argued to be key for managing difference and essential for effective stakeholder partnerships and shared decision-making.

### Arguments related to the purposes of PI in health and social care research

There was critical consensus (over 80% strongly agreed/agreed) at R1, across all stakeholder groups, that members of the public should be involved in publicly funded research, and are entitled to say what and how research is undertaken (see figure 2). There was also critical consensus that members of the public should be involved in research impacting on their own health or on National Health Survey functioning and clear consensus that they can influence how such research is used (see figure 2). However, over 60% agreed/strongly agreed that there are tensions between what members of the public and researchers see as the purpose of research and what constitutes good research.

At R2, we asked panellists to comment on whether they believed these/such tensions within the arena of health and social research could be resolved. Given that tensions between the different stakeholder groups in the PI enterprise have been highlighted elsewhere,[47] a somewhat unexpected finding from our data was critical consensus (75%) that tension resolution was possible. The percentage range across stakeholder groups was 67–89%. Individual stakeholder group responses are outlined in table 3.

Also unexpected was that, though tension was seen as an inevitable consequence of potential differences in perspective and/or expectations, it was not necessarily perceived as a negative issue. Rather it was seen by many as a means to stimulate critical debate.

> There should be tensions because that indicates passion; but the differences need to be managed in an open and transparent manner. [MP, R2]

**Table 2** Does it matter if stakeholder groups hold views considered biased by others?*

| Stakeholder group | n | Bias matters | Bias does not matter |
|---|---|---|---|
| Clinical academic researcher | 39 | 23 (59%) | 16 (41%) |
| Non-clinical academic researcher | 66 | 32 (48%) | 34 (52%) |
| Member of the public | 41 | 17 (41%) | 24 (59%) |
| Research manager/funder | 56 | 16 (29%) | 40 (71%) |
| Dual role | 27 | 10 (37%) | 17 (63%) |

*Data taken from R2.
R2, round 2.

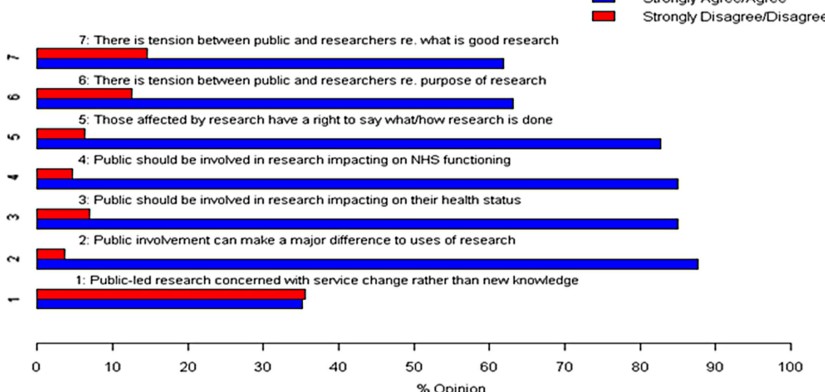

**Figure 2** What are the purposes of public involvement in health and social care research? Data taken from Round 1.

Tensions are a natural consequence of different stakeholders having different priorities. To some extent the debate they lead to could be regarded as conducive to better decisions. [NCA, R2]

This is something to value, rather than to be concerned about! As long as those considering the various perspectives understand in advance that there may be differences in opinion, then there aren't 'tensions', just differences. [CA, R2]

…if researchers, clinicians and members of the public work together at the start of a trial, it will significantly reduce these tensions. However I also think that you can never entirely remove the tensions, and to some extent this is a good thing as the element of struggle helps to refine the research question. [MP, R2]

Once again, the need to promote values of effective partnership working—including, for example, reciprocity, communication, reflexivity and learning from each other—was seen as a key to tension resolution over time.

There was support in the panellists' written responses at R1 for the idea that PI training for members of the public needed to not only include an overview of research design and methods but should also involve

education about political and policy context(s) which would lead to greater understanding of fundamental research drivers and processes. It was articulated by some panellists that training could also increase empowerment of members of the public, promoting and facilitating partnership working within the research team. However, panel members also reflected that this should not stop academic researchers from recognising and valuing the experiential knowledge of members of the public themselves.

### Arguments relating to the why and the how of PI

There was overall critical consensus (over 70% strongly agreed/agreed) that research is more ethical when members of the public are involved at all stages of the process. However, this statement was most strongly endorsed by members of the public (94% strongly agreed/agreed) and least endorsed by clinical (67%) and non-clinical (59%) academics. There was critical consensus across stakeholder groups (81% strongly agreed/agreed) that involvement is empowering for members of the public. Again, members of the public and those with multiple roles were much more likely to agree (75% strongly agreed/agreed) than were other groups (54%) that PI equalises the power between them and professionals. Clear consensus was also reached (over 65% agreed strongly/agreed) in relation to the statement that PI was often a 'tick the box' exercise, with members of the public more likely to express disappointment at the lack of opportunity to influence the research process. Some panellists (across all stakeholder groups) suggested that involvement was still often seen as being tokenistic, with 'lip service' only paid to the issue of PI.

There is the perception that it is an inconvenient tick box exercise giving no added value and actually slowing research process down. [CA, R2]

Until researchers recognise the importance and value of public involvement, and how it can improve the quality of their research, they are likely to give it low priority. [CA, R2]

**Table 3** Can tensions be resolved in health and social care research?*

| Stakeholder group | n | Tension resolution YES | Tension resolution NO |
|---|---|---|---|
| Clinical academic researcher | 38 | 34 (89%) | 4 (11%) |
| Non-clinical academic researcher | 66 | 44 (67%) | 22 (33%) |
| Member of the public | 41 | 35 (85%) | 6 (15%) |
| Research manager/funder | 54 | 37 (69%) | 17 (31%) |
| Dual role | 26 | 18 (69%) | 8 (31%) |

*Data taken from R2.
R2, round 2.

While PI is no longer optional it can still feel like a hassle for the academic community with gains that can seem remote. That needs to change otherwise people will continue being tokenistic. [NCA, R2]

Panellists voiced a number of concerns about involving members of the public in the research process, including the lack of funding to involve the public in the early stages of research question and proposal development. Time pressures related to grant applications and project-specific time-lines were also highlighted as problematic, including lack of time to take account for public members' illness and/or work commitments, the lack of realistic timeframes for reviews and, not least, the lack of time needed to develop and nurture research team relationships.

At R1, a number of panellists expressed the view that a 'fit-for-purpose model of PI, which reflected study and social contexts, was a more appropriate way of assuring effective PI than the various 'levels of PI' frameworks proposed in the PI literature.[48] Such frameworks were not popular with panellists, who argued that the hierarchical approach they embody to articulate different PI levels was misleading and not necessarily an indication of PI quality. The framework approach to PI was further criticised for not addressing all the facets of involvement, and as such having limited application and becoming potentially a driver for tokenism. Interestingly, INVOLVE[49] have recently updated their guidance to refer to 'approaches' rather than to 'levels' of involvement and also recognises that different approaches to PI might be used within the same project.

Following these responses, we asked panellists at R2 whether, in their opinion, they felt there were any circumstances where PI was inappropriate. There was critical consensus among the panellists that members of the public could be involved in all/any type of research—with the caveat that PI may be more challenging within the field of basic science research. The following observation made by one panellist, a RM/funder, serves to illustrate the potential for variation in the way members of the public can be involved:

…hard to imagine a rationale for no involvement, but it will vary from project to project. One argument is often made that there shouldn't be public involvement in pure basic biomedical research and certainly the level of involvement will be very different to that, say, of a clinical trial. But there is still potential for involvement for example in wider public debate/involvement about evaluating this type of research and making policy decisions about how much money to invest in this type of research than more applied research. Some scientific projects may require human samples and there is a role for the public in the governance issues around this type of research—or motivating others to donate tissue etc. I have come across some aspects of clinical trials where, for example, two different scales for measuring depression were compared—a very technical and limited bit of

work—with no room for influence from public involvement—but this was one part of a much larger trial where there was plenty of scope for involvement. So it may not always be appropriate for public involvement to influence the design and delivery of a project if it is very technical —but there is likely to be value of involvement in other aspects of the project around governance and funding decisions and perhaps dissemination in informing people that the research is going on—and explaining the implications of the research to a lay audience. [RM, Round 2]

## DISCUSSION
This article reports findings from an online, modified Delphi study involving a range of stakeholder groups to explore normative, substantive and process-related values underpinning PI in health and social care research. This approach to gathering experts' opinions and suggestions without the need to bring them together physically proved versatile and resource-efficient. In this study, Delphi panellists' responses were fairly evenly balanced across the various stakeholder groups; and the response rate was, in our view, acceptable given the 'cold call' approach.[50–52] Allowing panellists an opportunity to comment on their interpretation of the items and to express their views via open-question feedback increased the reliability of the study and improved the validity of the results.[42]

In the following discussion, we first consider key themes emerging from the data in relation to the normative, substantive and process-related values underpinning PI in health and social care research; second, we highlight the potential limitations of our research approach; and finally, we present our conclusions.

### Key themes
Our findings suggest high levels of self-reported expertise among members of the public involved in health and social care research, regardless of little 'formal' education and training in PI being reported. Panellists reported that timely and appropriate training is required by members of the public to support their involvement. Similarly, panellists also articulated that other actors in the research process require training in how to engage and support members of the public in their research roles. Support strategies suggested by panellists included, for example, advice and mentoring schemes.

Overall, our R1 survey identified critical consensus across stakeholder groups for many of the normative, substantive and process-related issues explored. These included that members of the public have an entitlement to be involved in the research process, including consideration of what research is undertaken and how it is used, that different types of knowledge are important and that members of the public have a unique knowledge. These areas of consensus highlight the extent to which PI is embedded in health and social care research. However, there were also areas of conflict by stakeholder group, for example, there was wide variation

in the percentages endorsing the normative position that research is more ethical where there is PI throughout. Although ethical concerns are paramount in the context of health and social care research, it is interesting to note that the two stakeholder groups least likely to endorse the importance of PI to ensure that research probity were CA and NCA. There was also disagreement about the extent to which PI was seen as equalising power between the public and professionals, and about the substantive issues around potential stakeholder group bias and representativeness. These concerns about the lack of representativeness of members of the public who participate in PI in research initiatives have preoccupied the PI community for some considerable time[53] [54] and it has been argued that lack of clarity about these concepts has led to PI being devalued.[53] That it remains a focus of concern to our panellists highlights the need for a more sustained debate around the nature of 'representativeness' and about implementing involvement that is meaningful and appropriate.

Most health and social care research areas, including preclinical, were deemed appropriate for PI, with many panellists articulating their views about how and when it might be feasible to introduce and evaluate involvement. This reflects recent changes in commitment by the UK funding bodies in incorporating PI across all research designs and at all stages of the research process.[5] Assessments about the appropriateness of PI need to be viewed within the overall opportunities for members of the public to influence project-specific agendas. A recent review of case studies conducted by Boote *et al*[17] highlighted potential tensions between different stakeholder groups involved in PI. However, in our study, the potential for tension was articulated as an inevitable consequence of collaborative working; individuals' agendas can be different when designing research studies. The acknowledgement and management of this standpoint was deemed essential in reconciling potential stakeholder difference. By promoting values of effective partnership, including, for example, communication, reflexivity and learning from each other, research relationships can develop over time, and tension can lessen as individuals work towards a common goal. The recognition that tensions are inevitable and can be used productively through questioning and discussion has also been highlighted by Abma and Widdershoven.[55]

It is interesting to note that although panellists recognised the normative debates around involving members of the public in health and social care research and acknowledged significant progress in doing so, they also reported shortcomings in process and substantive-related issues around the practice of PI in research, not least of which is the need for quality standards where involvement is embedded methodologically in research study designs. Promoting and evaluating PI as 'normalised' research practice was seen as a key in strengthening the evidence base around PI, which in turn, it was argued, would reduce the practice of tokenism.

## Delphi study limitations

In this study, we chose to adopt a modified Delphi approach for data collection. This involved the use of fixed choice and open questions in order to try to maximise our understanding of the issues under consideration. As with all survey approaches, there are inevitably limitations to the depth of the data obtained; and it could be important to follow-up key issues using more in-depth approaches, which would allow for more detailed exploration of less well understood and articulated issues.

Although Delphi techniques vary, face-to-face contact with participants at R1 has been found useful in increasing the response rate.[28] However, due to the size of our sample, many of the panellists were targeted 'cold' without any prior notice. This approach may have had an impact on our response rate at R1. The use of reminders is generally endorsed in texts on survey methods,[56] and in line with Peterson *et al*,[57] two reminder cover letters were emailed to non-responding participants at R1 and R2 of the survey to stimulate additional responses.

That our response rate to R1 was low was unsurprising given our approach; however, it was encouraging that a large percentage of responders to R1 subsequently completed R2. The Delphi technique requires continued commitment from participants throughout the data collection process. Consideration must be given to the fact that individual time constraints and lack of familiarity with the Delphi technique may have prevented some participants from being able to make such a commitment. Nonetheless, the quality of the responses provided made clear that those who did take part were firmly committed to offering us detailed and extremely thoughtful answers to our questions.

It could be argued that another potential limitation relates to the representativeness of our panellists—though as discussed above, the issue of what 'representativeness' involves in the context of PI remains a thorny one. Under half of those we approached participated in R1 of the study, and this proportion was further reduced at R2. Since those who opted in to the survey self-selected the stakeholder group with which they aligned themselves, we have no information about the groupings of those who opted out. We also have no information about other characteristics of interest for which we collected information from participants, for example, years of involvement in PI or extent of training in PI. We are therefore unable to comment meaningfully on the representativeness or otherwise of the study population. As previously clarified, we purposively set out to recruit PI experts to our study; it is likely, therefore, that those opting to take part were individuals with a particularly strong commitment to PI in research, who were therefore keen to endorse its validity. Conversely, our study may have failed to capture the views of people with experience, but not the strong commitment to or opinions about PI. Our findings may therefore offer an

overly optimistic picture. This needs to be borne in mind when interpreting the findings.

## CONCLUSIONS

This study has identified a number of key issues around PI implementation, in relation to the different weightings given by different stakeholder groups to the values around PI, and the resultant potential for conflict. This study serves as one of the few attempts to empirically gauge the views of key stakeholders on these issues. The findings suggest that there is not only a large degree of consensus across many of these issues, but also a healthy level of debate in some important areas. While acknowledging the problem of representativeness of our panellists, we feel that this Delphi survey provides us with important insight into the views of this group of PI experts.

Our findings also support the conclusion of Boote et al[58] that there is a need for best practice standards regarding PI, which research teams can follow when seeking to involve members of the public in the research process. The aim of the wider MRC research within which this Delphi study sits was to develop and pilot guidance to support assessment of the impacts of PI and to produce draft of the best practice standards. Findings from our modified Delphi study about commonly held PI values and about assessment of PI impacts (the latter is the subject of a separate article) have contributed to this wider work by identifying areas where conflict is likely to arise and suggesting ways such conflict can be negotiated so the PI agenda can move forward meaningfully.

**Author affiliations**
[1]Department of Public Health and Policy, University of Liverpool, Liverpool, UK
[2]Medicines for Children Research Network, University of Liverpool, Liverpool, UK
[3]University of Liverpool, Liverpool, UK
[4]Division of Health Research, Lancaster University, Lancaster, UK
[5]Institute for Health Research, Exeter Medical School, Exeter, UK

**Acknowledgements** The authors of this article are members of Public Involvement Impact Assessment Framework research team, which involves researchers at the Universities of Lancaster, Exeter and Liverpool. Other members of the PiiAF team who contributed to the Delphi Study were: Felix Gradinger, Elaine Hewis, Tim Rawcliffe and Paula Williamson. Gradinger reviewed and commented on the survey questionnaires in light of the literature review, he conducted as part of the wider MRC Study; Hewis and Rawcliffe took part in the expert workshops; Williamson contributed statistical expertise, including on the definition of consensus. The authors would also like to acknowledge the valuable input of the members of the Public Advisory Group connected to this study: Bert Green, Faith Harris-Golesworthy, Dina Lew, Irene McGill, Nigel Pyart; and members of the Advisory Network. They are indebted to all those who took part in the Delphi study and wish to thank participants for their time and insight.

**Contributors** DS was responsible for day-to-day management of the Delphi study, the qualitative data analysis and the drafting of the manuscript. JK was responsible for management and analysis of the quantitative data. JPr assisted in the expert workshops, in development of the survey questionnaires and commented on the manuscript. JP and NB contributed to

the conceptual development of the Delphi study and commented on the manuscript. MC, KF, FL and KW all commented on the survey documents and the manuscript. AG assisted with the expert workshops, in development of the questionnaires and commented on the manuscript. AJ had responsibility for the overall conceptual and methodological development of the Delphi study, supervision of DS and drafting/finalising of the manuscript.

**Funding** The study was supported by the Medical Research Council's Methodology Research Programme [G0902155/93948]

**Competing interests** JP was a member of the MRC Methodology Research Programme at the time this grant was awarded.

**Ethics approval** University of Liverpool Research Ethics Committee.

**Provenance and peer review** Not commissioned; externally peer reviewed.

**Data sharing statement** Exploring the scope for making the anonymised data files are available from the corresponding author at the University of Liverpool, which would provide a permanent, citable and open access home for the dataset.

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
