## [Reviewer comments · BMJ Open]

Some articles will have been accepted based in part or entirely on reviews undertaken for other BMJ Group journals. These will be reproduced where possible.

ARTICLE DETAILS

TITLE (PROVISIONAL)	Exploring areas of consensus and conflict around values underpinning public involvement in health and social care research: A modified Delphi study
AUTHORS	Snape, Dee; Kirkham, Jamie; Preston, Jennie; Popay, Jennie; Britten, Nicky; Collins, Michelle; Froggatt, Katherine; Gibson, Andy; Lobban, Fiona; Wyatt, Katrina; Jacoby, Ann

VERSION 1 - REVIEW

REVIEWER	Dr Jonathan Boote University of Sheffield, UK Member of advisory network for the study and a Delphi participant
REVIEW RETURNED	03-Nov-2013

GENERAL COMMENTS	The paper presents the findings of an interesting Delphi study on issues around the values underpinning public involvement in health research. I recommend that the journal accepts the paper subject to the authors address the following issues: 1. In the results section of the abstract, the sentence beginning, 'at an operational level' is unclear.2. In the conclusions edition of the abstract, I didn't understand the term 'value consensus'.3. As I understand it, the Delphi process/technique is not technically a survey due to the use of purposive and snowball sampling. So I would encourage the authors not to refer to a 'Delphi survey'. Where relevant, this should be changed to either 'Delphi process' or 'Delphi questionnaire'.4. In the lay summary, I didn't understand the final bullet-point.5. In the first paragraph of the introduction, I would recommend that the authors simply reference PCORI and the CPP, as the authors don't actually go on to discuss the approach to PI in these organisations in this paragraph.6. In the second paragraph of the introduction, the authors argue that an important challenge arises from the lack of an evidence base for PI but do not clearly set out the nature of this challenge.7. On page 8, I found the authors' discussion about how they defined clear and critical consensus to be very hard to follow, so I wonder if this could be set out more costly. Similarly, I found the paragraph following this discussion to be unclear.
--

	8. On page 9, the authors seem to be implying that training is a pre-requisite for involvement, and I wonder if this needs to be reconsidered/reworded. 9. On page 12, in the final quote from a participant, I wonder if the word 'extra' is missing after the word 'optional' 10. On page 14, the authors mention 'high' consensus. I wasn't sure what was meant by this term. 11. I wasn't sure I understood the first line in the conclusion. Could this be reworded?
--	---

REVIEWER	Sandy Oliver Institute of Education, University of London, UK
REVIEW RETURNED	08-Nov-2013

GENERAL COMMENTS	This paper reports a modified Delphi study exploring the values underpinning public involvement in health and social care research. They sought a range of people bring different perspectives on public involvement in research. This focus is important to remember. As it stands it is too easy for a reader to interpret the generic term 'stakeholder' as being a 'stakeholder in research' but in this context I think it is more accurately used to mean a 'stakeholder in public involvement in research'. For instance, the statement in line 21-22, page 2, might easily be understood as meaning that there is consensus across researchers and members of the public, rather than there being consensus across people from these two groups actively working together. The article summary, in line 2, page 3, makes this clear with a description of the panel being composed of PI experts. In line 10 this is described as a possible source of bias whereas I think it is the focus of the study, and that the study provides no insight into the values of people who do not engage in public involvement in research, whether they are researchers or members of the public. It may also have failed to capture the views of people with experience, but no strong views (either positive or negative) of public involvement in research. The issue of representativeness in public involvement appears in the abstract and in the lay summary (line 36, page 4) and the main text (line 47 page 14). This concept deserves greater attention given that methods required to involve people statistically representative of their peers are different from those required to involve people with skills to represent their peers on complex issues to professional groups. I am not convinced that Nilsen et al 2009 (page 5, line 13) provide evidence of public involvement being firmly rooted in UK research policy. The description of the panellists in the results section (line 29, page 25) could do with expanding. Describing the panel as a whole as having at least 5 years' experience of research and three quarters having responsibility for PI does not distinguish the expertise of researchers and the public in terms of public involvement in research. The details that follow are not clearly aligned with either researchers or members of the public, yet their meaning is very
---

	different for these two groups.
--	---------------------------------

REVIEWER	Sherine Gabriel Mayo clinic USA
REVIEW RETURNED	11-Nov-2013

- The reviewer completed the checklist but made no further comments.

VERSION 1 – AUTHOR RESPONSE

Reviewer 1 (Boote):

1. In abstract, results section, text re-written for clarity
2. In abstract, conclusions section, term 'value consensus' clarified
3. In methods, Dr Boote questions whether the term 'survey' is appropriate in the context of our Delphi Study. We would argue that it is, given various definitions of the term. The Oxford Dictionary defines a 'survey' as an investigation of 'the opinions or experience of a group of people by asking them questions' – which is what we did. More directly relevant, in the HTA review by McColl et al (Health Technology Assessment, 2001; 5: 31) on the design and use of questionnaires, the distinguishing features of the survey method are listed; and we are of the opinion that our work fits well with these features. In considering our position, we would also like to clarify to Dr Boote that this study did not use a 'snowballing' technique for sampling. We surveyed a self-selecting group of PI experts, identified from a wide range of sources as described in the Methods section.
4. In lay summary, wording of final bullet point revised accordingly.
5. In introduction, PCORI and CPP made reference to only, as requested.
6. In introduction, the nature of the challenge, as we see it, now clarified.
7. On P.8, description of clear and critical consensus re-written for clarification.
8. On P.9, point relating to training re-phrased as requested
9. On P.12, wording of quote checked and amended
10. On P.14, high consensus replaced by 'critical consensus'
11. In conclusion – first line re-worded for clarity, as requested.

Reviewer 2 (Oliver):

1. We thank Dr Oliver for the perceptive comment concerning the term 'stakeholder' and have re-worded the text where appropriate to clarify meaning.
2. We have added new text to the limitations section, to respond to the point made about the possible failure of the study to capture particular views.
3. We have added text and additional references about the issue of 'representativeness'
4. We have deleted reference to Nilsen et al, 2009 in the introduction
5. We have included additional analysis about the panellists, as requested new Table 1 and additional text in the Results section).

Reviewer 3 (Gabriel)

This reviewer had no specific comments.

We hope you will now feel able to accept the paper for publication and look forward to receiving a decision in due course.